# Direct Imaging of Radiation-Sensitive Organic Polymer-Based Nanocrystals at Sub-Ångström Resolution

**DOI:** 10.3390/nano14100872

**Published:** 2024-05-17

**Authors:** Elvio Carlino, Antonietta Taurino, Dritan Hasa, Dejan-Krešimir Bučar, Maurizio Polentarutti, Lidia E. Chinchilla, Josè J. Calvino Gamez

**Affiliations:** 1Istituto di Cristallografia del Consiglio Nazionale delle Ricerche (IC-CNR), 70126 Bari, Italy; 2Istituto per la Microelettronica e i Microsistemi del Consiglio Nazionale delle Ricerche (IMM-CNR), 73100 Lecce, Italy; antonietta.taurino@cnr.it; 3Department of Chemical and Pharmaceutical Sciences University of Trieste, 34127 Trieste, Italy; dhasa@units.it; 4Department of Chemistry, University College London, London WC1H 0AJ, UK; d.bucar@ucl.ac.uk; 5Elettra-Sincrotrone Trieste, Basovizza, 34149 Trieste, Italy; maurizio.polentarutti@elettra.eu; 6Departamento de Ciencia de los Materiales, Ingeniería Metalúrgica y Química Inorgánica, Facultad de Ciencias, Universidad de Cádiz, 11519 Puerto Real, Cádiz, Spain; lidia.chinchilla@uca.es (L.E.C.); jose.calvino@gm.uca.es (J.J.C.G.)

**Keywords:** radiation damage, polymers, soft matter, HoloTEM, atomic-resolution imaging, in-line holography, HRTEM

## Abstract

Seeing the atomic configuration of single organic nanoparticles at a sub-Å spatial resolution by transmission electron microscopy has been so far prevented by the high sensitivity of soft matter to radiation damage. This difficulty is related to the need to irradiate the particle with a total dose of a few electrons/Å^2^, not compatible with the electron beam density necessary to search the low-contrast nanoparticle, to control its drift, finely adjust the electron-optical conditions and particle orientation, and finally acquire an effective atomic-resolution image. On the other hand, the capability to study individual pristine nanoparticles, such as proteins, active pharmaceutical ingredients, and polymers, with peculiar sensitivity to the variation in the local structure, defects, and strain, would provide advancements in many fields, including materials science, medicine, biology, and pharmacology. Here, we report the direct sub-ångström-resolution imaging at room temperature of pristine unstained crystalline polymer-based nanoparticles. This result is obtained by combining low-dose in-line electron holography and phase-contrast imaging on state-of-the-art equipment, providing an effective tool for the quantitative sub-ångström imaging of soft matter.

## 1. Introduction

Transmission electron microscopy (TEM) methods have been extremely successful in the improvement in our understanding of the structural, electromagnetic, and chemical properties of matter, pushing the development of physics, materials science, and chemistry [1,2,3]. Richard Feynman stated the following in his famous lecture “Plenty of room at the bottom” in 1959: “*It would be very easy to make an analysis of any complicated chemical structure; all one would have to do would be to look at it and see where the atoms are. The only trouble is that the electron microscope is one hundred times too poor… I put this out as a challenge: Is there no way to make the electron microscope more powerful?*” [4]. Since then, research in electron microscopy, both on the technological and methodological sides, focused on overcoming the limitations to the spatial resolution power of the electron microscope [5,6,7], with the aim to enable the highest resolution and accuracy in the understanding of matter at the atomic level, for a conscious advance in a wide variety of fields of scientific and industrial relevance [1,2,3]. Nowadays, sub-ångström resolution is achievable by TEM and scanning TEM (STEM) imaging methodologies, owing mainly to the development of spherical aberration correctors [8] and high-coherence and high-brilliance field emission sources [9]. Furthermore, the recent progress in direct-conversion electron detectors enables atomic-resolution imaging with unsurpassed signal-to-noise ratios and acquisition speeds [10]. Unfortunately, however, the high-resolution TEM (HRTEM) imaging of soft matter at a sub-ångström resolution remains practically impossible [11,12,13] because of the severe and, so far, unavoidable specimen radiation damage occurring during the experiments [14]. Therefore, the biggest challenges in the development of TEM/STEM imaging and spectroscopies continue to be related to radiation damage, which is caused by the high density of electrons targeting the specimen [15]. For the typical HRTEM imaging of inorganic specimens, a density of electrons ρ of some thousands of e^−^/Å^2^ or more is delivered [16], while the study of organic matter requires extremely small electron doses to avoid structural damage. As a conservative rule of thumb, Henderson reports that, even at a cryogenic temperature, the electron density delivered to the biologic specimen should not exceed 5 e^−^/Å^2^ at a liquid nitrogen temperature and 20 e^−^/Å^2^ at a liquid helium temperature [17], which are orders of magnitude lower than those currently applied for the HRTEM imaging of single-particle inorganic materials, like metals or semiconductors, which are relatively robust to radiation damage. Furthermore, the higher the spatial resolution to be achieved, the higher the dose of electrons necessary for effective imaging [18]. A successful high-spatial-resolution imaging experiment should answer our need to know where the atoms in a structure (and which atoms) are but also, for example, if the bonds are locally strained or the structure is affected by a defect, as the properties of a material depend not only on the structure but are largely governed by strain and defects [19]. The steps necessary for effective HRTEM imaging experiments of single nanoparticles are numerous and mainly consist in finding, on the TEM grid, the nanoparticle well oriented along a zone axis, waiting until the drift of the specimen holder stops, accurately fine-tuning the electron optical conditions on the area of interest, namely, fine-focusing and illumination convergence angle adjustment (this latter step is mandatory to properly simulate the image to extract all the information encoded in the frame), and finally acquiring a high-signal-to-noise HRTEM image [2]. All these steps must be performed without damaging the nanoparticle structure and be repeated for a number of nanoparticles high enough to have statistical significance in addressing quantitatively the specimen properties.

This is why the sub-ångström HRTEM imaging of soft matter has not been reported so far in the literature despite several attempts [12,13]. Among soft matter, polymers are an example of extremely radiation-sensitive materials because electron densities lower than 0.2 e^−^/Å^2^ should be mandatory for imaging at a 1 Å resolution [18]. At these values of ρ, even the use of the most advanced high-sensitivity direct-conversion cameras results in images with a faint contrast, which are practically useless for quantitative measurements of the specimen properties. Furthermore, all the steps necessary before the acquisition of the HRTEM image might have already damaged the particle. Examples of the recent literature dealing with HRTEM on radiation-sensitive materials report the experimental difficulties in achieving effective imaging experiments even when they do not tackle the most sensitive case of small nanoparticles. For example, few-ångström-resolution imaging was obtained from large crystalline metal–organic framework specimens containing also heavy metals [13]. In other cases [12], the authors increased the spatial resolution in the HRTEM imaging of polymers, from 2.1 nm to 0.36 nm, by means of a special specimen preparation procedure which adds oxidants to the pristine materials.

The need to image small proteins, difficult to be crystallised for typical X-ray protein crystallography methods [20,21], at a high resolution was tackled by Cryo-EM, which was acknowledged with the Nobel Prize in chemistry in 2017 [22]. The method does not image but reconstruct, at a typical resolution which has, so far, ranged between about 4 Å and 2 Å, the three-dimensional structure of individual macromolecules, assumed to be in an identical or similar conformation, in specimens vitrified at cryogenic temperatures [23,24]. Cryo-EM requires the acquisition of numerous two-dimensional low-dose TEM images of macromolecules in different projections, which are then computationally combined to finally provide a three-dimensional reconstruction which can be interpreted as an electron density map. There are some limitations in the size of the protein particles that can be reconstructed, which should not exceed about 35 kDa, due mainly to the fact that the molecules are embedded in amorphous ice. Additionally, the models derived by Cryo-EM need to be validated, and large workgroups are committed to this aim, especially in the near-atomic resolution range [24]. A comprehensive review of the limits, performances, and ultimate perspectives in cryo-EM was provided by R. M. Glaeser [25].

Here is shown that the sub-ångström high-contrast direct imaging of pristine soft matter at room temperature is, conversely, possible by a recent experimental method that uses a combination of in-line electron holography and low-dose HRTEM, hence named HoloTEM. The main aim here is not to solve a specific material problem but make evident that soft matter can be imaged at the ultimate instrumental resolution by following a procedure capable of overcoming the experimental limitations encountered when imaging soft matter by HRTEM. The experiments were performed on advanced field emission gun spherical aberration-corrected TEM [26], equipped with high-speed, high-sensitivity direct-detection cameras. As a case study, we tackled a crystalline polymer-based material, highly sensitive to radiation damage. This approach is effective even on single nanometric pristine crystalline particles, and the relevant experimental atomic-resolution images can be quantitatively simulated, opening new perspectives in the study of radiation-sensitive materials.

In–line electron holography was established in the 1940s by Gabor [27] to overcome the spatial resolution limitations in TEM due to electron lens’ aberration. Since then, electron holography has been used for several different aims in electron microscopy (see [28] and references therein). The HoloTEM method uses extremely low-dose-rate real-time in-line holograms for the specimen survey, capable of detecting crystalline particles suitably oriented for HRTEM imaging. Once a particle has been found, the hologram enables one to safely check when the specimen drift stops. Then, the hologram is used for tuning the electron optical conditions by fine-focusing the objective lens and spreading the illumination conditions on the area of interest to minimise the electron dose while achieving the best illumination condition for a highly coherent electron wave ideal for HRTEM imaging. All these steps can be carried out while delivering a total dose well below the structural damage threshold [28]. As evidenced in the following, the possibility to use the low-dose hologram to evaluate in advance the diffracting conditions of the particles enables one to use doses of electrons for HRTEM imaging higher than the ones theoretically predicted [28]. The final low-dose HRTEM image is hence acquired using state-of-the-art high-speed high-sensitivity direct-detection imaging cameras that enable one to grab series of hundreds of low-dose HRTEM images to check offline the possible appearance of structural damage as a function of the density of electrons delivered to the specimen. The result of the experiments is a high-signal-to-noise-ratio HRTEM image, which is the sum of individual damage-free images whose spatial resolution, in our experiments, depends on the equipment set-up and, hence, could reach sub-ångström performances on state-of-the-art TEM instruments. It is worthwhile to remark that HoloTEM enables one to perform room-temperature atomic-resolution imaging on pristine organic materials without any staining or special specimen preparation procedures and can be applied not only to large specimens [13] but also to small, individual, not-identical nanoparticles, with their own crystal structure and polymorph statistical distribution within the material [29], enabling one to perform quantitative HRTEM experiments on soft matter like those possible so far only on radiation-robust inorganic nanomaterials.

## 2. Materials and Methods

In this study, two different polymers based on polyethylene glycol (PEG), a ubiquitous polymer with applications in a variety of fields like medicine, biology, and chemistry, were used in combination with caffeine (caf) and fluoroanthranilic acid (ana), as precursors for cocrystal synthesis in a polymer-assisted grinding (POLAG) process [30]. The discovery of a cocrystal between caf, 6-fluoroanthranilic acid (6Fana), and PEG-DME 1000 in our previous study prompted us to perform a cocrystal screen using other fluorinated anthranilic acid derivatives, which led to the discovery of a new cocrystal composed of caf, 5Fana, and PEG-DME 1000 (CAPeg). The new polymer-based cocrystals were prepared mechanochemically in the absence of liquid additives (see Appendix A). During the screening stage, we also observed that the replacement of PEG-DME with PEG-PPG produced an isostructural solid (CAP). The solid products obtained mechanochemically were preliminarily characterised using powder X-ray diffraction and differential scanning calorimetry. 

The powder X-ray diffraction (PXRD) patterns of the mechanochemically prepared solids were collected at room temperature using a Bruker D2 Phaser diffractometer (Bruker, Mannheim, Germany) equipped with a low-power (300 W) X-ray source (30 kV at 10 mA) generating Ni-filtered Cu *K*_a_ radiation (*λ* = 1.54184 Å) and an SSD160-2 detector. The steel sample holder had an internal volume of 300 µL, which could be reduced to 100 µL through a home-made cylindrical gearbox in polyvinylidene fluoride. The measurement parameters were 2*θ* angles from 5° to 35°, 2*θ* steps of 0.02°, and a counting time of 0.6 s per step. The measurements are reported in Appendix A and are compared with the simulated diffractograms (see Appendix A).

Differential scanning calorimetric (DSC) measurements were performed on a *Mettler Toledo DSC 3* instrument. Approximately 3 mg of each solid was weighed in a 40 μL alumina pan and covered with an alumina lid. The samples were heated under the flow of dry nitrogen gas from 30 °C to 250 °C, with a heating rate of 10 °C min^−1^. The DSC curves were processed using the *Mettler STAR^e^* data evaluation software (version 16.40). The relevant measurements are reported in Appendix A.

Single crystals of CAPeg were obtained through recrystallisation from the melt (see Appendix A) and studied at the XRD1 beamline at the Elettra Synchrotron to determine the material’s crystallographic structure (see Appendix A). Standard low-dose TEM/STEM experiments performed on pristine materials at room temperature immediately resulted in the disruption of the polymeric matter, already during the survey necessary to find the particles of interest. This evidence urged the use of a different approach for TEM investigation. As demonstrated here, HoloTEM enabled us to achieve the ultimate resolution in low-dose HRTEM experiments while avoiding specimen damage. Details on the TEM specimen’s preparation and the state-of-the-art equipment used for the HoloTEM experiments are reported in Appendix A, respectively. 

In-line electron holography (see Figure 1) enabled us to detect, with high-contrast, the nanoparticles of the polymeric matter sustained on a standard TEM Cu grid covered with a thin carbon film, while delivering an extremely small density of electrons: ρ ≈ 0.1 e^−^/Å^2^.

The ray diagram in Figure 1 points out that, by tuning the electron optics, an in-line hologram (magnified ten times in Figure 1 for reader convenience) is formed in the back focal plane of the objective lens, where the interference between the reference wave and the wave diffused by the specimen makes the nanoparticle sharply visible despite the low-dose, with the latter being insufficient for a standard multibeam imaging live survey as formed on the image plane (see Figure 1). The experimental hologram and the multibeam image shown in Figure 1 are representative of what is actually seen by the scientist in real time, at a density rate of current ≤ 1.2 e^−^/s Å^2^, during the survey to find the area of interest on a standard Cu (C-coated) grid by live low-dose in-line holography or standard live low-dose imaging, respectively. 

The nanoparticles imaged in Figure 1 are cocrystals made of caffein, 5-fluoranthranilic acid, and polyethylene-polypropylene copolymer (CAP) that were dispersed on the TEM grid (see below and Appendix A.). The peculiarities of the HoloTEM method for the specimen survey are evident by comparing the in-line hologram and the relevant TEM image in terms of contrast (see red lines’ intensity profiles on the hologram and on the image) and field of view. The high contrast of the hologram (about 33%) compared to the faint contrast of the conventional multibeam image (less than 4%), points out that, according to Rose’s criterion [31], the particles are practically invisible in a standard imaging survey at a low dose, whereas the holograms ensure immediate and good particle detection (see also Appendix A). Furthermore, the field of view of the in-line hologram is thousands of times wider than the one of the standard images, enabling a realistic search for the particles of interest across a standard TEM specimen that can contain particles of different sizes and kinds (see Appendix A). It is worth further underlining that the HRTEM image in Figure 1 is representative of the conditions of low-dose live particle search and not the HRTEM image that can be acquired by the HoloTEM procedure. The difference between the two conditions is immediately evident when comparing the low-contrast HRTEM in Figure 1 with the high-contrast HRTEM image on the same particle, as shown in Appendix A. The latter has a contrast that enables sub-Å-resolution imaging on equipment like the one used in this work, as it can be derived from the relationship between the fluence, the contrast, and the attainable resolution according to the Rose criterion [14,28,30]. The data in Figure 1 also represent a synopsis of a HoloTEM experiment: using the real-time hologram at a low dose and a low dose rate, the particle can be located, choosing those well oriented for HRTEM imaging. The electron optics can then be optimised, as detailed below, and the specimen holder drift can be checked, until it stops. Hence, the set-up can rapidly be switched to the HRTEM mode, acquiring 100–200 images in the same area, with an exposure time for each image on the millisecond scale (see Appendix A). This procedure ensures a safe density of current ρ < 2 e^−^/Å^2^ for each image and enables full control over the appearance and eventual development of radiation damage (see Appendix A). Finally, the images can be summed to obtain a final HRTEM micrograph with a good signal-to-noise ratio that can be numerically simulated to quantify the specimen properties (see Appendix A).

The crystallographic structure of the material derived by the synchrotron XRD experiments was used in the quantitative simulations of the atomic-resolution imaging experiments here. Further details and examples are reported in Appendix A. HoloTEM combines the use of in-line holograms with low-dose HRTEM, resulting in an experimental procedure which enables an extremely accurate control of the equipment illumination system and an effective way to detect the particle of interest, with diffraction conditions suitable for HRTEM imaging, on a standard unstained TEM specimen, while monitoring the possible radiation damage [28]. It should be remarked that, in a conventional HRTEM experiment for single-nanoparticle imaging, the dose delivered during the exposure time for image acquisition is only a fraction of the problem [13]. The greater issue is due to the irradiation to which the specimen is exposed while finding the particles of interest on the TEM grid. At the values of ρ necessary not to damage the specimen, it is impossible, in a standard imaging experiment, to distinguish any organic unstained particle due to the low scattering power of the light elements [18,28] (see Figure 1). Further specimen irradiation occurs while waiting until the mechanical inertia of the specimen holder ends, during particle orientation and during the fine-tuning of electron optics in the region of interest. All these operations are mandatory before imaging acquisition for a quantitative analysis and can be properly performed by HoloTEM [16]. 

## 3. Results and Discussion

The results reported in the following show representative HoloTEM experiments on polymer-based specimens that provide evidence that sub-ångström HRTEM images on soft matter can be obtained with an unprecedent confidence on all the experimental parameters necessary for the quantitative HRTEM imaging of single pristine nanoparticles, on the dose reaching the specimen, and on its effect.

Figure 1 shows the raw HRTEM image of a CAP particle, together with the relevant hologram (see the inset), and demonstrates that high-quality low-dose HRTEM images can be obtained on a pristine polymeric material. 

The HRTEM image is the sum of 100 images, where each image received a ρ~7 e^−^/Å^2^ without showing detectable damage (see Materials and Methods and Appendix A). The experiments point out that monitoring the possible damage by HoloTEM procedures enables one to maximise the electron dose delivered to the specimen before the onset of structural damage. In fact, the expected dose limits [17] can be properly relaxed when a particle is well oriented along a zone axis; a very low dose rate is used [1,30,32,33]; protective layers of hydrocarbon contaminants are formed [14,33,34]; the illumination spread is maximised; and the possible damage development is monitored by multiple low-dose image acquisition (see §6, Appendix A). During the experiments we noticed that the particle disruption is strongly related to its diffraction conditions. A reason that can contribute to this behaviour is that, when the particle is well oriented along a zone axis, the scattering is eminently elastic [35], whereas the damage to the crystal structure is mainly due to inelastic scattering [36], the latter being reduced in electron-channelling conditions [37]. Leaving unchanged all the other experimental parameters, this leads to a higher robustness to the damage of well-oriented particles with respect to particles not oriented for high-channelling conditions (see Appendix A), even if, according to the knowledge up until now, the effect of the channelling alone could not explain quantitatively the observed phenomenon. To our knowledge, the role of the particle diffraction conditions has not been considered so far when calculating the maximum resolution reachable in TEM imaging before damage [14,28]. Hence, the values reported in the literature represent the minimum in the electron dose deliverable to a particle irrespective of its diffraction condition. On the contrary, higher dose values could be safely delivered when a particle is well oriented along a zone axis, enabling one to achieve a higher spatial resolution. Low-dose live in-line holograms enable a conscious choice of particle orientation during a specimen survey. A systematic study on tailored specimens would be necessary to quantify this aspect, but the results in Appendix A evidence how the dose delivered to a well-oriented crystalline polymer particle can be at least two orders of magnitude higher than previously believed. Indeed, for reader convenience, we report in Appendix A an example of an irradiation experiment on a pristine unstained particle of pure PEG, where a density of electrons up to at least 10^2^ e^−^/Å^2^ does not cause the evidence of particle damage, and a density of electrons of 85.4 × 10^3^ e^−^/Å^2^ is necessary to make most of the particle volume amorphous.

The high signal-to-noise ratio of the experimental HRTEM image obtained by HoloTEM, which is, at a first glance, from Figure 1, comparable to the one achievable in inorganic matter, enables us to quantify the material’s properties by measuring the lattice spacing of each nanoparticle forming a cluster, as shown by the diffractogram in Figure 1b and the relevant list of the measured lattice spacings. It is worth noting that the cluster shape immediately highlights the tendency of polymer-based cocrystals to form closed structures while maintaining mechanical flexibility, as already observed in our previous studies [30].

Figure 2a is the raw HRTEM image of a 12 nm × 18 nm thin nanoparticle of a CAPeg cocrystal, isostructural to a CAP cocrystal (see Materials and Methods and Appendix A). The micrograph is the sum of 100 images. Each image has been exposed to a density of electrons ρ = 2 e^−^/Å^2^. The structure of CAPeg, as derived by the synchrotron XRD experiments (see Figure 2c below, Appendix A, and the associated content CIF structural file), was used to simulate the HRTEM results (see Figure 2b and the inset marked in pale blue in Figure 2a) by a JEMS computer program using multislice full dynamical calculations [38]. The orientation of the particle with respect to the microscope optical axis was addressed by comparing the experimental diffractograms with those calculated by JEMS using the structural CIF file obtained by the XRD experiments (see the CIF file supplied in Supplementary Information). The simulation marked by the pale blue square within the raw experimental HRTEM image is calculated for the [3, 6, $\bar{4}$] zone axis of the triclinic crystal cell and well matches the experimental contrast. It is worthwhile to remark that the influence of the mutual transfer function (MTF) of the detector (see the manual of JEMS [38]) would rescale and, hence, decreases the simulated intensity, making the agreement between the experimental and simulated images even better. The effect of the MTF has to be mandatory considered in the simulation, for example, if the aim is to solve the crystal structure from the HRTEM, but not for the purpose of this study, where the structure is known from the XRD experiments. In Figure 2, the experimental diffractogram (d), the simulated diffraction pattern (e), and the measured (d_exp_) and calculated (d_theo_) lattice spacings are also reported. The HRTEM image and the relevant diffractogram evidence that the particle is a crystalline monodomain. The structure and size of these nanoparticles are directly accessed by HoloTEM atomic-resolution imaging experiments and related to the synthesis conditions [30], demonstrating that HRTEM imaging and relevant simulations can be performed on polymeric materials, despite their sensitivity to radiation damage.

Moreover, further subtle structural features at a sub-Å spatial resolution can be quantitatively achieved on the specimen, as shown in Figure 3. Figure 3a is a raw HRTEM image showing part of a large thin foil of CAPeg in the [10, 3, 2] zone axis. The experimental image is the sum of 100 frames. Each frame has been exposed to ρ = 0.8 e^−^/Å^2^ to check offline for possible damage (see Appendix A). The analysis of Figure 3 is helpful for understanding some subtle peculiarities of the cocrystal system studied. Indeed, the HRTEM images of CAPeg have some uncommon features, mostly evident on relatively large foils some hundreds of nm in size, which could be related to the typical flexibility of polymers, well known at a macroscopic scale and here directly evidenced at an atomic scale. In fact, we observed small in-plane and out-of-plane tilts making the lattice fringes’ spacing slightly different from place to place. Some areas showed compressive strain and others tensile strain; the distortion was hardly visible without comparing the experimental image with a reference (see Appendix A). Interestingly, such local strain did not generate visible extended defects and cracks, as it would have been the case in other, more rigid solids [40]. These results demonstrate that the study of crystalline polymeric materials at atomic resolutions allowed by HoloTEM would very likely require the modification of some of the paradigms we are used to, including the concept of rigidity often associated with the concept of crystal, where, when the crystal symmetry is broken by stress, it results in regions bounded by extended defects [40].

Figure 3b shows the diffractogram of the HRTEM image in (a). The intensities are distributed on rings, and their anisotropic distribution on the rings indicates that the foil consists of grains with the same orientation in the direction normal to the foil and textured in the plane of the foil [41]. Some of the most visible spacings have been numbered from 1 to 11 in the diffractogram in Figure 3b, and the relevant experimental values are reported together with Miller’s indexes and the spacing calculated by JEMS. The blue square in (a) marks an individual grain shown at a high magnification in Figure 3c, where fine structural details can be distinguished but, to be quantified and rationalised, need comparison with the pertinent image simulation [42]. The relevant diffractogram is shown in Figure 3d, where the (0, −2, 3) and (1, −2, −2) reflections are marked. The relevant vectors are the base vectors for the plane perpendicular to the [10, 3, 2] direction, which is parallel to the one of the primary beam. The comparison between the experimental and simulated diffractogram is reported in Appendix A, where the experimental pattern is superimposed onto the simulated one for ease of comparison. In the diffractogram, the reflections (4, −6, −11) and (3, −10, 0) are also marked by circles, as they correspond to two distinguished sub-Å spacings, pointing that the highest spatial resolution enabled by the equipment can be reached here on soft matter (see Appendix A), despite its sensitivity to radiation damage. It is worthwhile to remark that dynamical scattering could result in non-linear effects producing sub-Å intensities in the diffractogram that could not correspond to real structural features within the particles, hence not contributing to the image’s resolution [43,44]. A way to disclose this subtle point is to simulate the HRTEM image and identify the structural features generating the measured sub-Å spacing. In the experimental image in Figure 3c, this task is a bit complicated by the complexity of the atom arrangement within the unit cell, the distortion in the bonds, and some thickness variations within the observed area. Indeed, the fitting image simulation superimposed in the top-left part of Figure 3c has been calculated for a thickness of 27 nm, whereas the area in the centre of the image has been simulated for a thickness of 18 nm. The details of the image simulation are reported in Appendix A. Figure 3e is the crystal cell of CAPeg as calculated from the synchrotron X-ray diffraction measurements, shown in the same orientation [10, 3, 2] as the experimental HRTEM image. The grey rectangle marks part of the atomic configuration of the CAPeg crystal cell that is shown magnified in the lower-right part of the figure. The display has been obtained using the Mercury software [39]. The atoms in the cell are shown with different colours: the H atoms are green; the F atoms are yellow; the O atoms are red; the C atoms are grey; and the N atoms are blue. The pale-red lines in Figure 3e are the traces of planes (3, −10, 0) and (4, −6, −11). These lattice planes are those generating the intensities marked in the experimental diffractogram, corresponding to the measured spacings of 85 pm and 93 pm, respectively. In the lower-right part of Figure 3e, these spacings are marked with the dashed grey and red lines, respectively. This result evidences that the spacing at 93 pm is due mainly to the marked carbon–oxygen dumbbell, whereas the spacing at 85 pm is due mainly to the marked carbon–carbon atoms. The same atomic configuration has been superimposed for reader convenience onto the image simulation in Figure 3c to underline their structural correspondence. It should be marked that not only the distances but also the angles formed by the dumbbell in the (4, −6, −11) and (3, −10, 0) planes are coherent between the experiments and the simulations, as shown in Figure 3, pointing out the structural correspondence and ruling out that the sub-Å spacing in the diffractogram in Figure 3e could have been due to a non-linear effect in the HRTEM image.

## 4. Conclusions

The HoloTEM experiments demonstrated that pristine polymeric nanomaterials could be imaged at a sub-ångström resolution at room temperature. Up until now, it was believed to be impossible to directly study soft matter at this degree of spatial resolution and accuracy and compare quantitatively an experimental atomic-resolution image to the geometry of a simulated one to understand the subtle properties of these materials. Here, the direct comparison between experiments and simulations using full dynamical calculations was enabled by the relatively high signal-to-noise ratio of the experimental images, despite the low scattering power of the organic materials, and it was made possible by the accurate control of both the electron optical and particle scattering conditions achievable by the hologram at a dose below the threshold for structural damage. To our knowledge, the direct comparison between atomic-resolution HRTEM experiments and image simulations had not been previously reported in the literature for soft matter. Furthermore, the above experiments pointed out that the well-controlled imaging conditions allowed by HoloTEM, the possibility to recognise in advance, through the holograms, the channelling conditions suitable for HRTEM, and the use of a low dose rate of electrons, coupled with the formation of a very thin, protective hydrocarbon layer, enabled the use of a total density of electrons higher than the one previously believed possible for the imaging of soft matter [14,28]. Direct imaging at a sub-ångström resolution opens new scenarios for understanding the properties of soft matter, and, as recently theoretically predicted, it is key for the application of in-line electron holography to the atomic-resolution three-dimensional shape reconstruction of crystalline soft matter nanoparticle from a single projection [45]. In the case of CryoEM, the accuracy of HRTEM imaging and the relevant simulations enabled by HoloTEM can be used to improve the knowledge and constraints necessary for accurate and reliable CryoEM reconstruction. Among the other aspects which HoloTEM opens in the study of soft matter, there is the development that can be foreseen from the quantitative understanding of the phase shift information contained in the holograms [46].

## Data Availability

Experimental data other than those published and included in the Appendix A can be provided upon motivated requests to the corresponding author.

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
