# Peer review of "Direct Imaging of Radiation-Sensitive Organic Polymer-Based Nanocrystals at Sub-Ångström Resolution"

_nanomaterials, 2024, doi:10.3390/nano14100872_

Round 1

Reviewer 1 Report

Comments and Suggestions for Authors

Dear editors, dear authors,

the presented manuscript reports on a study of beam sensitive polymer-based nanocrystals with sub-Angstrom resolution
using a method called HoloTEM, that was introduced by the authors in an earlier publication.
The method seems to be a clever way to properly prepare the particle prior to acquisition with a very low electron dose.
Overall the paper is for my personal taste a bit too much focused on advertisement of the method and too less on the results.
However, I can understand that the method should be promoted, as it enables imaging of polymer nanocrystals at atomic scale.
The paper is well readable and interpretations are deduced stringently from the result. So in principle I recommend the
paper for publication after some revisions

- I think that the HoloTEM method should be briefly explained in a paragraph, so that a "TEM-reader" roughly knows what is done
during the experiment without reading the original paper (switching from diffraction to imaging mode, focusing of image, etc.)
- "Unfortunately, however, HRTEM..." is this really meant like this?
- I think: "Manheim" -> "Mannheim"
- the unit of the volume of the sample holder should be l not l^3
- I did not get what the DSC measurements are used for in the paper?  They only are in the supplementary infos.
- Why is it important to state that the images are magnified ten times in Scheme 1. It is a scheme and the actual
 magnification will anyway change depending on the post-magnification (e.g. on the screen).
- page 10: The experimental contrast was compared the simulation. However, it is well known that there
  was the so called Stobbs factor problem in contrast comparison between TEM experiment and simulation.
  Nothing was said in the paper about the MTF of the used camera, which is certainly necessary to get a comparable image and simulation
- page 11: "To our knowledge direct...the literature" -> One should certainly add that this sentence holds only for polymer-type of materials.
- in the suppl. materials: Block -> Bloch

Author Response

Answer to reviewer 1

Dear reviewer, thank you for considering our paper and for your positive evaluation. In the following the point-to-point answers to your questions:

Q:- I think that the HoloTEM method should be briefly explained in a paragraph, so that a "TEMreader" roughly knows what is done during the experiment without reading the original paper (switching from diffraction to imaging mode, focusing of image, etc.)

A: We agree with the reviewer, it is always difficult to apply a new methodology learned from a research paper and we indeed dedicated many efforts to report in this paper the details necessary for a microscopist to reproduce our experiments; this is done starting from the Materials and Methods section and keep going through Results and Discussion section and further complemented in the Supplementary Information document. Nevertheless, we recommend the reader also go into the reference [28].

Q: - "Unfortunately, however, HRTEM..." is this really meant like this?

A: Yes. As matter of the fact, this statement follows the report of the capability of modern equipment of imaging at sub-Å resolution that, unfortunately, cannot be straightforwardly applied to soft-matter due to radiation damage.

Q:- I think: "Manheim" -> "Mannheim"

A: thank you, we amended the typo in the revised version of the manuscript.

Q:- the unit of the volume of the sample holder should be l not l^3

A: Thank you, we amended the typo in the revised version of the manuscript.

Q:- I did not get what the DSC measurements are used for in the paper? They only are in the

supplementary infos.

A: We agree with the reviewer, the DSC are not used in the paper; nevertheless, we decided to give to the reader this further information on the studied system in the section of the supplementary information.

Q:- Why is it important to state that the images are magnified ten times in Scheme 1. It is a scheme and the actual magnification will anyway change depending on the post magnification (e.g. on the screen).

A: The aim of scheme 1 is also, as reported in the paper, to be a kind of synopsis of the actual survey process by HoloTEM, and we decided to magnify ten times the hologram in the Scheme 1 for reader convenience, as this is the typical magnification available in the standard binocular of a TEM with respect to what is observed on the microscope usual phosphorus screen where we see a much wider field of view.

Q:- page 10: The experimental contrast was compared the simulation. However, it is well known that there was the so called Stobbs factor problem in contrast comparison between TEM experiment and simulation. Nothing was said in the paper about the MTF of the used camera, which is certainly necessary to get a comparable image and simulation

A: we agree with the reviewer that the MTF has to be mandatory taken into account for a simulation of the image intensity when, for example, HRTEM images are used to derive the concentration of a species in a III-V semiconductor alloy or to derive quantitatively the intensity of the exit wave-function and this was indeed the driving force for the work of Hÿtch and Stobbs at the beginning of the nineties of the last century. Again, an attempt to derive the crystal structure of an alloy from the HRTEM image would definitely require taking into account the origin of the mismatch between the intensity of experimental and simulated HRTEM images (Stobb’s factor). As known, the lower intensity in the experimental image can be reasonable reproduced taking into account the mutual transfer function of the detector. This is unnecessary in our case as what we were interested to demonstrate, was not the agreement of the overall intensity of the HRTEM experimental and simulated images for structural determination. Indeed, we preferred to solve the crystal structure by XRD synchrotron experiments to avoid further complication to the already complicated matter we are dealing with here. Here we use the HRTEM simulation to evidence the capability of HoloTEM to provide sub-Å structural resolution that was believed unreachable on soft matter, and for this purpose we need to compare not the absolute intensity of the simulated image but the geometrical distribution of the intensity related to the projected potential of the specimen. We added in the revised version of the paper a statement to mention this aspect. We believe that the HRTEM result by HoloTEM could indeed be used for the determination of an unknown structure but this task cannot be deepened here and will be tackled in a dedicated future work.

Q:- page 11: "To our knowledge direct...the literature" -> One should certainly add that this sentence holds only for polymer-type of materials.

A: Thank you, we added the relevant statement pointing that we deal with soft matter

Q:- in the suppl. materials: Block -> Bloch

A: Thank you, we amended the typo in the revised version of the Supplementary materials

Reviewer 2 Report

Comments and Suggestions for Authors

The paper titled "Direct Imaging of Radiation Sensitive Organic Polymer-Based Nanocrystals at Sub-Ångström Resolution" by Elvio Carlino et al. and collaborators showcases the groundbreaking potential of the HoloTEM method in imaging pristine crystalline polymer-based single nanoparticles at sub-angstrom resolution while mitigating radiation damage. This method, which combines electron in-line holography and phase contrast imaging, offers a significant advancement in the field, enabling room-temperature imaging of nanoparticles with exceptional spatial resolution and a high signal-to-noise ratio.

The prospect of overcoming radiation damage and achieving high signal-to-noise ratios will undoubtedly excite cryo-EM scientists. However, it's worth noting that the test nanoparticle used in this study exhibits exceptional quality, which may not fully represent the complexity of real-world samples. Nonetheless, the paper is well-written, and the results are robust. I recommend publication after addressing my minor concerns.

1.       While the study focuses on a specific polymer-based material, it would be valuable to explore the performance of the HoloTEM method across a broader range of soft matter samples to assess its versatility and applicability.

2.       Comparative Analysis: A direct comparison with established techniques such as Cryo-EM could provide a more comprehensive understanding of the strengths and limitations of the HoloTEM method, enhancing its contextual relevance within the scientific community.

3.       Incorporating a more detailed statistical analysis of the imaging results would bolster the credibility of the conclusions drawn from the data, providing a more robust foundation for the study's findings.

4.       Clarity in Scheme Representation: The depiction of Scheme 1 appears to be challenging to comprehend. I recommend that the authors provide additional detail, particularly on the right part of the scheme, to enhance its clarity and aid in understanding for readers.

Author Response

Answer to reviewer 2

Dear reviewer, thank you for considering our paper and for your positive evaluation. In the following the point-to-point answers to your questions:

Q: 1. While the study focuses on a specific polymer-based material, it would be valuable to

explore the performance of the HoloTEM method across a broader range of soft matter

samples to assess its versatility and applicability.

A: We definitely agree that HoloTEM should now be applied to other soft-matter materials system, and we already did some experiments on proteins whose results will be the subject of forthcoming publications.

Q: 2. Comparative Analysis: A direct comparison with established techniques such as Cryo-

EM could provide a more comprehensive understanding of the strengths and limitations of

the HoloTEM method, enhancing its contextual relevance within the scientific community.

A: We agree with the reviewer and, as reported in the paper, the accuracy achievable in the imaging by HoloTEM could be really of help in improving the structure reconstruction achievable by Cryo-EM and also open the way for further synergies that need to be explored.

Q: 3. Incorporating a more detailed statistical analysis of the imaging results would bolster

the credibility of the conclusions drawn from the data, providing a more robust foundation for the study's findings.

A: as reported in the paper, the results reported are representative of hundreds of particles studied by HoloTEM.

Q: 4. Clarity in Scheme Representation: The depiction of Scheme 1 appears to be

challenging to comprehend. I recommend that the authors provide additional detail,

particularly on the right part of the scheme, to enhance its clarity and aid in understanding

for readers.

A: We added a statement in the caption of Scheme 1 to further clarify its aim.